# Factors associated with post-pandemic acceptance of COVID-19 vaccines among students in three Nigerian universities

**Adaobi Uchenna Mosanya**[1], **Adaeze Ezekwelu**[1], **Ezinwanne Jane Ugochukwu**[1], **Blessing Onyinye Ukoha-Kalu**[2]*

1 Clinical Pharmacy and Pharmacy Management, Faculty of Pharmaceutical Sciences, University of Nigeria, Nsukka, Enugu State, Nigeria, 2 School of Medicine, University of Nottingham, Nottingham, England, United Kingdom

* blessing.ukoha-kalu@nottingham.ac.uk

**Data Availability Statement:** All relevant data are within the paper and its supporting information files.

## Abstract

### Background

The COVID-19 pandemic impacted the world in every aspect. Higher institutions were greatly affected because the outbreak disrupted the teaching and learning structure. Vaccines decrease the rate of infection and transmission of the virus, but the presence of some myths has led to hesitancy towards the vaccine.

### Objective

The purpose of the survey was to assess the knowledge, perception, and acceptance of the COVID-19 vaccine among undergraduate students in Enugu State, Nigeria.

### Methods

This was a cross-sectional study carried out among undergraduate students at the University of Nigeria Nsukka (UNN), the Institute of Management and Technology (IMT), and Enugu State University of Technology Enugu state (ESUT), Nigeria between March and November 2023. These institutions were chosen based on their large student populations, diverse academic offerings, and significant geographical coverage within the state. Data collection was done using a 26-item validated self-administered questionnaire. Statistical Product and Service Solutions (SPSS) version 25 with appropriate descriptive (frequency and percentage) and inferential statistics (Chi-square) were used to analyze the data.

### Results

1,143 completed questionnaires were obtained. The modal age range was 18–24 years accounting for 814 (71.2%) of the participants. A total of 577 (50.5%) participants demonstrated a good level of knowledge while 685 (59.9%) showed a positive perception of the COVID-19 vaccine. Gender (p = 0.010) and institution (p < 0.001) were associated with their perception of the COVID-19 vaccine. In addition, knowledge and perception of the COVID-

**Funding:** The author(s) received no specific funding for this work.

**Competing interests:** The authors have no competing interests relevant to this article's content.

19 vaccine were significantly associated with its acceptance at p = 0.038 and < 0.001, respectively.

## Conclusion

This study reveals that COVID-19 vaccine acceptance among university students in Enugu State, Nigeria, remains low despite moderate knowledge and generally positive attitudes, with perceptions playing a more significant role than knowledge. The findings highlight the need for educational interventions that not only provide accurate information but also actively address misconceptions. To improve vaccine uptake, public health campaigns should focus on shifting perceptions through culturally sensitive, institution-specific strategies.

## Introduction

COVID-19 was first recorded in December 2019 with the discovery of a cluster of patients suffering from pneumonia of unclear cause in Wuhan City, Hubei province of China. The severe acute respiratory syndrome coronavirus- 2 which has spread to more than 200 countries with more than 760 million reported cases and 6.9 million confirmed deaths [1] is a new strain from the family of coronaviruses (Coronaviruses) identified as the cause of these uncommon infections [2].

As of June 2023, over 13 billion vaccine doses have been administered [1]. Although WHO announced the end of the COVID-19 emergency phase in May 2023 [1], there is a continuous rise in the number of new cases globally [3]. This highlights the need for continuous vaccination. In reality, vaccine hesitancy is defined as "the delay in acceptance or refusal of vaccination despite the availability of vaccination services" [4] and is a significant issue in Africa today [5]. A good proportion of Nigerians just like in other countries, considered the vaccine safe and effective in the prevention and control of the pandemic [6], however, the acceptance rate ranges from 20–56% [7].

Tertiary institutions, globally, were highly affected by the COVID-19 pandemic [8–10] resulting in psychological distress, interruptions in learning, and anxiety. Previous studies assessing COVID-19 vaccine acceptance or hesitancy as well as the contributing factors were not specifically carried out among the students of tertiary institutions. There seems to be a mix in the population, introducing a high level of heterogeneity in the generated data. Some compelling factors were identified as reasons for vaccine hesitancy such as conspiracy theories, disbelief, uncertainty about its safety, etc, but none highlighted the factors peculiar to students of higher institutions. Enugu state is one of the states in Nigeria that has more than 30 tertiary institutions. Therefore, there is a need to assess the acceptance of the COVID-19 vaccine among students of tertiary institutions in Enugu state and associated factors. This would provide data specific to students of higher institutions and form the basis for generalization. Additionally, there is a potential to raise awareness and foster more positive attitudes toward vaccination. Identifying the factors that affect vaccine acceptance among this population would assist in designing and implementing a program aimed at improving vaccine uptake in Enugu State.

## Methods

### Study design

This was a cross-sectional study carried out among students of three higher institutions in Enugu state Nigeria. The ethical approval to carry out the study was granted by the Research Ethics Committee of the Faculty of Pharmaceutical Sciences, University of Nigeria with a reference number FPSRA/UNN/22/0044. During the study, no personal identifying data was collected to ensure the confidentiality of the responses. We obtained informed written consent from the study participants before enrolling them in the study.

### Study setting

The study was carried out at three higher institutions in Enugu State, Nigeria. These were the University of Nigeria Nsukka (UNN), the Institute of Management and Technology (IMT), and the Enugu State University of Technology Enugu state (ESUT), Nigeria between March and November 2023. These institutions were chosen based on their large student populations, diverse academic offerings, and significant geographical coverage within the state. We prioritized institutions with large and diverse student bodies to ensure that our sample would be representative of the broader student population in Enugu State. To capture a range of perspectives and experiences, we selected institutions that differ in their academic focus and type (a federal university, a state university, and a polytechnic). The selected institutions are strategically located in different parts of Enugu State, ensuring that the study captures data from students in various urban and rural settings.

### Eligibility criteria

The eligibility criteria for participation in the study were being an undergraduate student in the above higher institutions and granting informed written consent to take part.

### Sampling and sample size estimation

This study employed a non-probability purposive and convenience sampling technique to recruit the participants. With the assumption of a marginal error of not more than 2% with a 95% confidence level and a prevalence of COVID-19 vaccine acceptance of approximately 20%, the sample size was estimated as 1537 [11].

### Data collection instrument

A 26-item self-administered questionnaire was used for this study (S1 File). Content validation was carried out by three academic lecturers of the Department of Clinical Pharmacy and Pharmacy Management, Faculty of Pharmaceutical Sciences, University of Nigeria Nsukka. Section A of the questionnaire consists of eight items on the sociodemographic characteristics of the participants. Section B consists of eight questions that assess the knowledge of the participants regarding the COVID-19 vaccine with a "Yes" or "No" response option. Section C consists of eight perception items regarding the COVID-19 vaccine with a five-point Likert scale response from Strongly disagree- Strongly agree. Finally, two questions were used to assess the acceptance level of the participants towards the COVID-19 vaccine with a "Yes" or "No" or "I don't know" response option. The questionnaire was distributed to the students in various settings such as their hostels, in their classrooms before lectures, or in public spaces within the institution's premises.

## Data analysis

The completed questionnaires were coded and entered into Microsoft Excel. Thereafter, the raw data were cleaned and then exported into the Statistical Product and Service Solutions (SPSS) version 25 (IBM Corp., Armonk, N.Y., USA) for analysis. Frequencies and percentages were used to describe the analyzed data. Inferential statistics were done by computing the Chi-square test of the association of knowledge, perception, and acceptance scores and the demographic characteristics of the respondents, and the p-value was set as 0.05. After computing the knowledge and perception response scores, it was found that they were not normally distributed. Therefore, the median was used as the cut-off. The respondents with scores equal or more than the median score had good knowledge while those having scores equal or above the median were categorized as having a positive perception of the COVID-19 vaccine.

## Results

In Table 1, a total number of 1,143 students completed the survey while the estimated sample size was 1537. Therefore, the response rate was 74%. Most of the participants 814 (71.2%) were within the age range of 18–24 years. A total of 610 (53.4%) of the students live off-campus.

**Table 1. Demographic information of respondents (N = 1,143).**

| Demographics | | Number | Percentage |
|---|---|---|---|
| **Age (Years)** | Less than 18 | 51 | 4.5 |
| | 18–24 | 814 | 71.2 |
| | 25–31 | 269 | 23.4 |
| | 32–38 | 8 | 0.7 |
| | Above 38 | 3 | 0.3 |
| **Residence** | Hostel | 533 | 46.6 |
| | Off campus | 610 | 53.4 |
| **Gender** | Male | 318 | 27.8 |
| | Female | 825 | 72.2 |
| **Marital status** | Single | 1074 | 94.0 |
| | Married | 69 | 6.0 |
| **Mode of admission** | Post-UTME | 1054 | 92.2 |
| | Direct- Entry | 89 | 7.8 |
| **Religion** | Christianity | 1114 | 97.5 |
| | Islam | 19 | 1.7 |
| | African traditional religion | 10 | 0.9 |
| **Ethnic group** | Igbo | 1079 | 94.4 |
| | Yoruba | 25 | 2.2 |
| | Hausa | 10 | .9 |
| | Cameroon | 3 | .3 |
| | Annang | 4 | .3 |
| | Ghanaian | 2 | .2 |
| | Ibibio | 7 | .6 |
| | Edo | 4 | .3 |
| | Idoma | 8 | .7 |
| | others | 1 | .1 |
| **Higher institution** | University of Nigeria | 538 | 47.1 |
| | Institute of Management and Technology | 279 | 24.4 |
| | Enugu State University of Science and Technology | 326 | 28.5 |

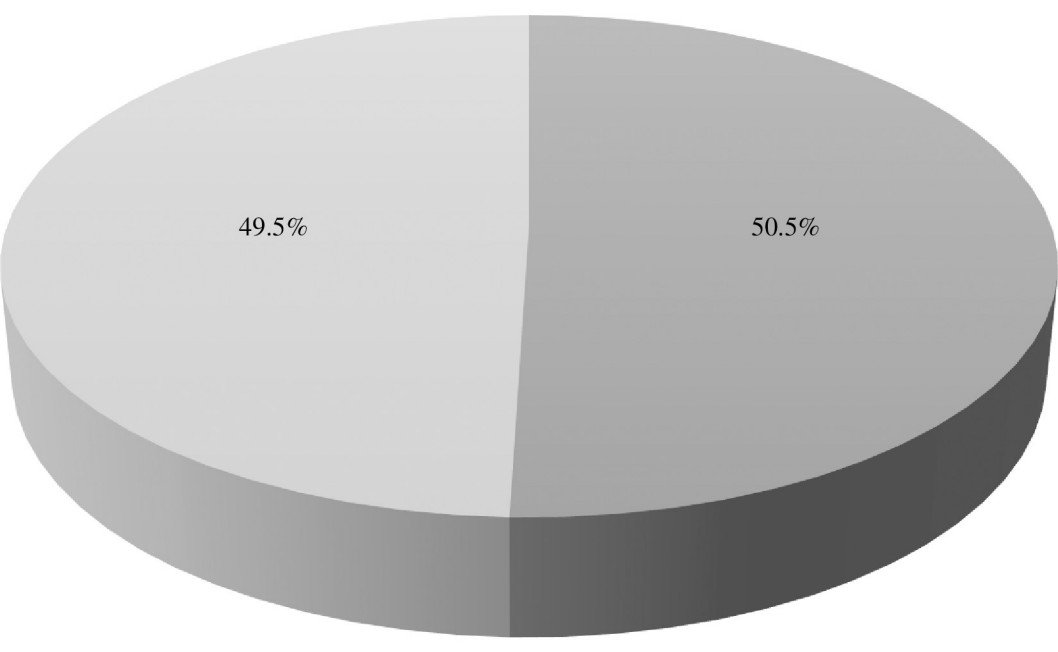

■ Good Knowledge   ■ Poor Knowledge

**Fig 1. The level of knowledge of the COVID-19 vaccine.**

Most of the students, 1054 (92.2%) gained admission through the post-Unified Tertiary Matriculation Examination (UTME). A total of 577 (50.5%) of the respondents had a good knowledge of the COVID-19 vaccine (Fig 1). In Table 2, the statement "Is it dangerous to use an overdose of COVID-19 vaccine?" had the highest percentage of "Yes" answers 1031 (90.2%). Also, 585 (51.2%) of the respondents stated "No" to the statement "Is COVID-19 vaccine suitable for pregnant women?" From the study, 94.0% had not taken the COVID-19 vaccine. The level of acceptance was very low 69 (6%) and only about 5% of those who had taken it would recommend it to family or friends. In Table 3, almost half of 478 (41.9%) agreed that

**Table 2. Knowledge and acceptance of COVID-19 vaccine.**

| S/N | Statements | No | | Yes | |
|---|---|---|---|---|---|
| | | Number | Percentage | Number | Percentage |
| 1 | Is the COVID-19 vaccine given by only injection | 228 | 19.9 | 915 | 80.1 |
| 2 | Can the COVID-19 vaccine protect the receiver from getting COVID-19 infection? | 291 | 25.5 | 852 | 74.5 |
| 3 | Is it dangerous to use an overdose of COVID-19 vaccine? | 112 | 9.8 | 1031 | 90.2 |
| 4 | Does COVID-19 vaccination cause allergic reactions? | 294 | 25.7 | 849 | 74.3 |
| 5 | Does vaccination increase autoimmune disease? | 365 | 31.9 | 778 | 68.1 |
| 6 | Is the COVID-19 vaccine suitable for pregnant women? | 585 | 51.2 | 558 | 48.8 |
| 7 | Can everyone including children receive the COVID-19 vaccine? | 330 | 28.9 | 813 | 71.1 |
| 8 | Do COVID-19 vaccines have side effects? | 225 | 19.7 | 918 | 80.3 |
| 9 | Have you taken the vaccine? | 1074 | 94.0 | 69 | 6.0 |
| 10 | If yes, do you encourage your family/friends/relatives to get vaccinated? | 8 | 0.7 | 55 | 4.8 |

Table 3. Perception towards COVID-19 vaccine.

| S/N | Statements | Strongly Disagree | | Disagree | | Neutral | | Agree | | Strongly Agree | |
|---|---|---|---|---|---|---|---|---|---|---|---|
| | | n | % | n | % | n | % | n | % | n | % |
| 1 | COVID-19 vaccine is safe | 48 | 4.2 | 85 | 7.4 | 371 | 32.5 | 454 | 39.7 | 185 | 16.2 |
| 2 | COVID-19 Vaccine is essential for use | 66 | 5.8 | 125 | 10.9 | 347 | 30.4 | 454 | 39.7 | 151 | 13.2 |
| 3 | COVID-19 may cause infection | 128 | 11.2 | 406 | 35.5 | 362 | 31.7 | 160 | 14.0 | 87 | 7.6 |
| 4 | COVID-19 vaccine may not be effective | 89 | 7.8 | 188 | 16.4 | 341 | 29.8 | 415 | 36.3 | 110 | 9.6 |
| 5 | It is not possible to reduce the incidence of COVID-19 without vaccination | 156 | 13.6 | 247 | 21.6 | 241 | 21.1 | 330 | 28.9 | 169 | 14.8 |
| 6 | Scary information about the vaccine is rampant on social media | 22 | 1.9 | 49 | 4.3 | 139 | 12.2 | 478 | 41.8 | 455 | 39.8 |
| 7 | COVID-19 vaccine can protect me from getting infected | 117 | 10.2 | 150 | 13.1 | 394 | 34.5 | 330 | 28.9 | 152 | 13.3 |
| 8 | I am afraid to take the COVID-19 vaccine | 89 | 7.8 | 100 | 8.7 | 340 | 29.7 | 336 | 29.4 | 278 | 24.3 |

there are "Scary information about the vaccine is rampant on social media". Also, 454 (39.7%) of the respondents agreed that "COVID-19 vaccine is safe and essential for use". Table 4 shows the proportion of study participants who had good knowledge, good perception, and acceptance of the COVID-19 vaccine. Tables 5 and 6 shows that there was no significant association ($p > 0.05$) between the sociodemographic variables and the knowledge about the COVID-19 vaccine. However, the perception of the respondents towards the COVID-19 vaccine was significantly associated with gender ($p = 0.010$) and institution ($p < 0.001$). In addition, knowledge and perception of the COVID-19 vaccine were significantly associated with its acceptance at $p = 0.038$ and $< 0.001$ respectively. Table 7 shows the logistic regression results. Only knowledge and perception regarding the vaccine were significantly associated with acceptance of the COVID-19 vaccine after the chi-square analysis. Only these two variables were entered as covariates in the logistics analysis to identify the predictors. They both emerged as predictors after adjusting for the confounding effects of both. Those with good knowledge have less odds of accepting the vaccine than those with poor knowledge. However, those with good perception have four times the odds of accepting the vaccine than those with poor perception.

## Discussion

This study's findings evaluated the level of knowledge, perception, and acceptance of the COVID-19 vaccine among students in three tertiary institutions in Enugu state, Nigeria. About half of the students had a good knowledge of the COVID-19 vaccine. This is comparable to a study by Jiang et al showing good knowledge about the COVID-19 vaccine among the participants [12]. One may assume that such an observation was because undergraduate students are expected to be exposed to certain information inaccessible to the other less privileged members of the general population. This current study identified a significant association between good knowledge and vaccine acceptance. A significantly lower proportion of those with good knowledge who accepted the vaccine than those with poor knowledge. The logistics regression showed that those with good knowledge were less likely to accept the vaccine than those with poor knowledge. Perhaps, knowledge about the vaccine is not necessary to accept the vaccine.

On the other hand, good perception was a highly predictive factor for vaccine acceptance. Perhaps, it is sufficient for the population to have a good perception of the vaccine to accept vaccination. This is in line with the outcomes from a scoping review done on COVID-19 vaccine hesitancy in Africa. It was found that among other factors that facilitate the acceptability of the COVID-19 vaccine were knowledge and positive perception about the vaccine [5].

**Table 4. Good knowledge, good perception and acceptance of COVID-19 vaccine.**

| Characteristics | | Total (1143) | Good knowledge (577) | Good Perception (685) | Acceptance (69) |
|---|---|---|---|---|---|
| Age (Years) | Less than 18 | 51(4.5) | 27(52.9) | 29(56.9) | 3(5.9) |
| | 18–24 | 814(71.2) | 415(51.0) | 481(59.1) | 47(5.8) |
| | 25–31 | 267(23.4) | 129(48.3) | 166(62.2) | 16(6.0) |
| | 32–38 | 8(0.7) | 5(62.5) | 6(75) | 2(25) |
| | Above 38 | 3(0.3) | 1(33.3) | 3(100) | 1(33.3) |
| Residence | Hostel | 533(46.6) | 270(50.7) | 321(60.2) | 27(5.1) |
| | Off campus | 610(53.4) | 307(50.3) | 364(59.7) | 42(6.9) |
| Gender | Male | 318(27.8) | 155(48.7) | 171(53.8) | 21(6.6) |
| | Female | 825(72.2) | 422(51.2) | 514(62.3) | 48(5.8) |
| Marital Status | Single | 1074(94) | 545(50.7) | 642(59.8) | 65(6.1) |
| | Married | 69(6.0) | 32(46.4) | 43(62.3) | 4(5.8) |
| Mode of Admission | JAMB | 1054(92.2) | 540(51.2) | 623(59.1) | 65(6.2) |
| | DE | 89(7.8) | 37(41.6) | 62(69.7) | 4(4.5) |
| Religion | Christianity | 1114(97.5) | 564(50.6) | 666(59.8) | 65(5.8) |
| | Islam | 19(1.7) | 10(52.6) | 14(73.7) | 3(15.8) |
| | ATR | 10(0.9) | 3(30.0) | 5(50.0) | 1(10.0) |
| Ethnicity | Igbo | 1079(94.4) | 548(50.8) | 651(60.3) | 62(5.7) |
| | Yoruba | 25(2.2) | 11(44.0) | 14(56.0) | 1(4.0) |
| | Hausa | 10(0.9) | 3(30.0) | 5(50.0) | 2(20.0) |
| | Cameron | 3(0.3) | 1(33.3) | 0(0.0) | 0(0.0) |
| | Annang | 4(0.3) | 2(50.0) | 3(75.0) | 0(0.0) |
| | Ghanaian | 2(0.2) | 2(100) | 1(50.0) | 1(50.0) |
| | Ibibio | 7(0.6) | 2(28.6) | 6(85.7) | 1(14.3) |
| | Edo | 4(0.3) | 4(100) | 1(25.0) | 1(25.0) |
| | Idoma | 8(0.7) | 4(50.0) | 3(37.5) | 1(12.5) |
| | Others | 1(0.1) | 0(0.0) | 3(100) | 0(0.0) |
| Higher Institution | UNN | 538(47.1) | 285(53.0) | 293(54.5) | 29(5.4) |
| | IMT | 279(24.4) | 128(45.9) | 162(58.1) | 25(9.0) |
| | ESUT | 326(28.5) | 164(50.3) | 230(70.6) | 15(4.6) |
| Taken the Vaccine | No | 1074(94.0) | 551(51.3) | 627(58.4) | NA |
| | Yes | 69(6.0) | 26(37.7) | 58(84.1) | NA |

In this study, gender and institution of the study participants were significantly associated with their acceptance towards the COVID-19 vaccine with females showing a higher proportion with good perception as compared to the males. In addition, perception was significantly associated with acceptance of the vaccine. This would serve as a guide in targeting the males through campaigns aimed to increase vaccine uptake. However, it is interesting to note that a similar study in Ontario provided a contrary conclusion. Females were found to have poor perception of the vaccine and higher hesitancy compared to the males [13]. On the other hand, the institutions with low proportions of the participants having good perceptions would be targeted for improved vaccine uptake.

The findings from this study should be interpreted in the light of its strengths and limitations. One of its strengths was the focus on only students of tertiary institutions unlike other previous studies in Nigeria which included a heterogeneous population [14–16]. This would allow highlighting implications for policies geared towards increased COVID-19 vaccine acceptance and uptake. However, the limitation of using a questionnaire that was only face-validated would pose a problem in the generalizability of the findings.

**Table 5. Association between Sociodemographic and level of knowledge, perception, and acceptance of the respondents.**

| Characteristics | | Knowledge (A) N (%) | | χ2 | Perception (B) N (%) | | χ2 | Acceptance (C) N (%) | | χ2 |
|---|---|---|---|---|---|---|---|---|---|---|
| | | Good knowledge | Poor Knowledge | | Good Perception | Poor Perception | | Acceptance | Rejection | |
| **Age (Years)** P(A) = .823 P(B) = .439 P(C) = .058 | Less than 18 | 27 (2.4) | 24(2.1) | 1.522 | 29(2.5) | 22(1.9) | 3.760 | 3(0.3) | 48(4.2) | 9.115 |
| | 18–24 | 415(36.3) | 399(34.9) | | 481(42.1) | 333(29.1) | | 47(4.1) | 767(67.1) | |
| | 25–31 | 129(11.3) | 138(12.1) | | 166(14.5) | 101(8.8) | | 16(23.2) | 251(22.0) | |
| | 32–38 | 5(0.4) | 3(0.3) | | 6(0.5) | 2(0.2) | | 2(0.2) | 6(0.5) | |
| | Above 38 | 1(0.1) | 2(0.2) | | 3(0.3) | 0(0.0) | | 1(0.1) | 2(0.2) | |
| **Residence** P(A) = .959* P(B) = .897* P(C) = .244* | Hostel | 270(23.6) | 263(23.0) | 0.003* | 321(28.1) | 212(18.5) | 0.017* | 27(2.4) | 506(44.3) | **1.355*** |
| | Off-campus | 307(26.9) | 303(26.5) | | 364(31.8) | 246(21.5) | | 42(3.7) | 568(49.7) | |
| **Gender** P(A) = .507* **P(B) = .010*** P(C) = .718* | Male | 155(26.9) | 163(28.8) | 0.441* | 171(15.0) | 147(12.9) | **6.603*** | 21(1.8) | 297(26.0) | 0.130* |
| | Female | 422(36.9) | 403(35.3) | | 514(45.0) | 311(27.2) | | 48(4.2) | 777(68.0) | |
| **Marital Status** P(A) = .562* P(B) = .771* P(C) = 1.000* | Single | 545(47.7) | 529(46.3) | 0.336* | 642(56.2) | 432(37.8) | 0.085* | 65(5.7) | 1009(88.3) | <0.001* |
| | Married | 32(2.8) | 37(3.2) | | 43(3.8) | 26(2.3) | | 4(0.3) | 65(5.7) | |

*Continuity correction value for 2 X 2 tables P(A). P(B) and P(C) stands for the p-value for the knowledge, perception, and acceptance of the COVID-19 vaccine chi-square association with the respective variables.

**Table 6. Association between sociodemographic and level of knowledge, perception and acceptance of the respondents.**

| Characteristics | | Knowledge (A) N (%) | | χ2 | Perception (B) N (%) | | χ2 | Acceptance (C) N (%) | | χ2 |
|---|---|---|---|---|---|---|---|---|---|---|
| | | Good knowledge | Poor Knowledge | | Good Perception | Poor Perception | | Acceptance | Rejection | |
| **Mode of Admission** P(A) = .101* P(B) = .066* P(C) = .686* | JAMB | 540(47.2) | 514(45.0) | 2.690* | 623(54.5) | 431(37.7) | 3.380* | 65(5.7) | 989(86.5) | .164* |
| | DE | 37(3.2) | 52(4.5) | | 62(5.4) | 27(2.4) | | 4(0.3) | 85(7.4) | |
| **Religion** P(A) = .423 P(B) = .383 P(C) = .170 | Christianity | 564(49.8) | 550(48.1) | 1.723 | 666(58.3) | 448(39.8) | 1.917 | 65(5.7) | 1049(91.8) | 3.543 |
| | Islam | 10(0.9) | 9(0.8) | | 14(1.2) | 5(0.4) | | 3(0.3) | 16(1.4) | |
| | ATR | 3(0.3) | 7(0.6) | | 5(0.4) | 5(0.4) | | 1(0.1) | 9(0.8) | |
| **Ethnicity** P(A) = .294 P(B) = .219 P(C) = .089 | Igbo | 548(47.9) | 531(46.5) | 10.742 | 651(57.0) | 428(37.4) | 11.907 | 62(89.0) | 1017(89.0) | 15.074 |
| | Yoruba | 11(1.0) | 14(1.2) | | 14(1.2) | 11(1.0) | | 1(0.1) | 24(2.1) | |
| | Hausa | 3(0.3) | 7(0.6) | | 5(0.4) | 5(0.4) | | 2(0.2) | 8(0.7) | |
| | Annang | 2(0.2) | 2(0.2) | | 3(0.3) | 1(0.1) | | 0(0.0) | 4(0.3) | |
| | Ghanaian | 2(0.2) | 0(0.0) | | 1(0.1) | 1(0.1) | | 1(0.1) | 1(0.1) | |
| | Ibibio | 2(0.2) | 5(0.4) | | 6(0.5) | 1(0.1) | | 1(0.1) | 6(0.5) | |
| | Edo | 4(0.3) | 0(0.0) | | 1(0.1) | 3(0.3) | | 1(0.1) | 3(0.3) | |
| | Idoma | 4(0.3) | 4(0.3) | | 3(0.3) | 5(0.4) | | 1(0.1) | 7(0.6) | |
| | Others | 3(0.3) | 3(0.3) | | 3(0.3) | 2(0.2) | | 1(0.1) | 5(0.5) | |
| **Higher Institution** P(A) = .157 **P(B) = < .001** P(C) = .055 | UNN | 285(24.9) | 253(22.1) | 3.706 | 293(25.6) | 245(21.4) | **22.422** | 29(2.5) | 509(44.5) | 5.785 |
| | IMT | 128(11.2) | 151(13.2) | | 162(14.2) | 117(10.2) | | 25(2.2) | 254(22.2) | |
| | ESUT | 164(14.3) | 162(14.2) | | 230(20.1) | 96(8.4) | | 15(1.3) | 311(27.2) | |
| **Taken the Vaccine** *P(A) = < **0.038*** P(B) = < **0.001*** | No Yes | 551 (48.2) 26 (2.3) | 523 (45.8) 43 (3.8) | **4.283*** | 627(54.9) 58(5.1) | 447(39.1) 11(1.0) | **16.749*** | NA | NA | |

NA: Not applicable because taking the vaccine was interpreted as acceptance

**Table 7. Logistics regression COVID-19 vaccine predictors.**

| Variables | Acceptance of COVID-19 vaccine n (%) | COR (95% CI) | AOR (95% CI) |
|---|---|---|---|
| Knowledge | | | |
| Poor (566) | 43 (62.3) | Reference | Reference |
| Good (577) | 26 (37.7) | 0.574 (0.348–0.948) * | 0.595 (0.359–0.986) * |
| Perception | | | |
| Poor (458) | 11 (15.9) | Reference | Reference |
| Good (685) | 58 (84.1) | 3.759 (1.951–7.243) ** | 3.692 (1.914–7.119) ** |

*p< 0.05

** p < 0.001

Nevertheless, conclusions can be made regarding the acceptance of the COVID-19 vaccine among students of tertiary institutions in Enugu State, Nigeria. Firstly, there was a moderate level of good knowledge about the vaccine. Secondly, the acceptance level was very poor. Thirdly, perception and knowledge were significant predictors of vaccine acceptance. Therefore, implications for future research should be done using a content-validated questionnaire in addition to face validation. Also, to improve the generalizability of the findings, efforts should be directed to increasing the response rate. In addition, the study design may adopt a case-control method for more robust conclusions.

An implication for policy is a targeted campaign to improve the perception concerning the COVID-19 vaccine in all tertiary institutions which would increase the vaccine acceptance and uptake. The campaigns can be in the form of seminars, workshops, etc. organized specifically for students of tertiary institutions in Nigeria.

## Conclusion

This study provides valuable insights into the post-pandemic acceptance of COVID-19 vaccines among university students in Enugu State, Nigeria. Despite a moderate level of knowledge and generally positive perceptions towards the vaccine, acceptance rates remained low, influenced significantly by the students' perceptions rather than their knowledge alone. The findings underscore the importance of targeted educational interventions that not only disseminate accurate information but also actively address misconceptions and enhance positive perceptions. This approach could be instrumental in improving vaccine uptake among this demographic. Our study highlights the critical role of perception in vaccine acceptance, suggesting that future public health campaigns should prioritize shifting perceptions through culturally sensitive, institution-specific strategies. These findings contribute to the ongoing dialogue on vaccine hesitancy and suggest actionable steps for increasing vaccine acceptance in similar contexts.

## Policy implication

There is a need to ensure that information concerning the COVID-19 vaccine should be spread to all institutions via health promotion campaigns and public health talks. The heads of tertiary institutions should organize seminars, and workshops on vaccination acceptance and recommendations to all students.

## Supporting information

**S1 File. Study questionnaire.**
(PDF)

## Acknowledgments

The authors wish to acknowledge the students who participated in this study.

## Author Contributions

**Conceptualization:** Adaobi Uchenna Mosanya, Blessing Onyinye Ukoha-Kalu.

**Data curation:** Adaobi Uchenna Mosanya, Blessing Onyinye Ukoha-Kalu.

**Formal analysis:** Adaobi Uchenna Mosanya, Ezinwanne Jane Ugochukwu, Blessing Onyinye Ukoha-Kalu.

**Investigation:** Adaobi Uchenna Mosanya, Adaeze Ezekwelu, Ezinwanne Jane Ugochukwu, Blessing Onyinye Ukoha-Kalu.

**Methodology:** Adaobi Uchenna Mosanya, Adaeze Ezekwelu, Blessing Onyinye Ukoha-Kalu.

**Resources:** Adaobi Uchenna Mosanya, Blessing Onyinye Ukoha-Kalu.

**Supervision:** Blessing Onyinye Ukoha-Kalu.

**Writing – original draft:** Adaeze Ezekwelu, Blessing Onyinye Ukoha-Kalu.

**Writing – review & editing:** Adaobi Uchenna Mosanya, Adaeze Ezekwelu, Ezinwanne Jane Ugochukwu, Blessing Onyinye Ukoha-Kalu.

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
