## [Decision Letter · Decision Letter 0]

3 Jul 2024

PONE-D-24-14109Factors associated with post-pandemic acceptance of COVID-19 vaccines among students in three Nigerian UniversitiesPLOS ONE

Dear Dr. Ukoha-kalu,

Thank you for submitting your manuscript to PLOS ONE. After careful consideration, we feel that it has merit but does not fully meet PLOS ONE’s publication criteria as it currently stands. Therefore, we invite you to submit a revised version of the manuscript that addresses the points raised during the review process.

We look forward to receiving your revised manuscript.

Kind regards,

David Chibuike Ikwuka, Ph.D.

Academic Editor

PLOS ONE

Reviewers' comments:

Reviewer's Responses to Questions

**Comments to the Author**

1. Is the manuscript technically sound, and do the data support the conclusions?

Reviewer #1: Partly

Reviewer #2: Yes

Reviewer #3: Yes

2. Has the statistical analysis been performed appropriately and rigorously? 

Reviewer #1: Yes

Reviewer #2: No

Reviewer #3: Yes

3. Have the authors made all data underlying the findings in their manuscript fully available?

Reviewer #1: Yes

Reviewer #2: Yes

Reviewer #3: Yes

4. Is the manuscript presented in an intelligible fashion and written in standard English?

Reviewer #1: Yes

Reviewer #2: Yes

Reviewer #3: Yes

5. Review Comments to the Author

Reviewer #1: The authors have studied an important issue that continues to plaque control of infectious diseases through vaccination in particular, and for a no less important condition as COVID-19.

Background: The authors need to review other local studies on this subject and situate them in the background to the study and discussion. (Amuzie CI, Odini F, Kalu KU, Izuka M, Nwamoh U, Emma-Ukaegbu U et al. COVID-19 vaccine hesitancy among healthcare workers and its socio-demographic determinants in Abia State, Southeastern Nigeria: a cross-sectional study. Pan Afr Med J. 2021 Sep 3;40: 10.; COVID-19 vaccine hesitancy and willingness to pay: emergent factors from a cross-sectional study in Nigeria. Vaccine X. 2021 Dec;9: 100112; Njoga EO, Mshelbwala PP, Abah KO, Awoyomi OJ, Wangdi K, Pewan SB et al. COVID-19 vaccine hesitancy and determinants of acceptance among healthcare workers, academics and tertiary students in Nigeria. Vaccines (Basel). 2022 Apr 15;10(4): 626.; Amuzie CI, Odini F, Kalu KU, Izuka M, Nwamoh U, Emma-Ukaegbu U et al. COVID-19 vaccine hesitancy among healthcare workers and its socio-demographic determinants in Abia State, Southeastern Nigeria: a cross-sectional study. Pan Afr Med J. 2021 Sep 3;40: 10; Uzochukwu IC, Eleje GU, Nwankwo CH, Chukwuma GO, Uzuke CA, Uzochukwu CE et al. COVID-19 vaccine hesitancy among staff and students in a Nigerian tertiary educational institution. Ther Adv Infect Dis. 2021 Nov 1;8: 20499361211054923.)

Methods: What constitutes good knowledge and perception needs to be outlined.

Results: The p-values in the tables should be placed in a separate column, not under the characteristics. Percentages of teh gender under knowledge added up to 1279%. It is not clear hwy this should be so.

Discussion:

Finally, the discussion needs depth. Results from this study and other referenced studies were merely quoted with little attempt to critically appraise the findings and situate them within the context of this study and referenced works. Feeble attempt was made to provide plausible explanation for most findings from this study.

Conclusion: The authors have left out the main finding form this study, low COVID-19 uptake. Modifiable factors that affect this needs to be mentioned, as a way of recommendation.

Others: Few edits and comments are made in the attached manuscript.

Reviewer #2: INTRODUCTION

The introduction provides a concise overview of the COVID-19 pandemic, its impact, and the importance of vaccines. The purpose of the study is well-defined and aligned with the background information. The introduction provides context about the pandemic in Nigeria and the significance of vaccine acceptance among university students. However, the introduction covers a broad range of topics, including the pandemic's global impact, vaccine hesitancy, and Nigeria's vaccination efforts. While relevant, some information could be omitted or summarized to maintain focus. The introduction presents numerous statistics, which, although important, may overwhelm the reader. Some statistics could be saved for the results section or summarized in a table. The introduction jumps abruptly from discussing the pandemic's impact to the study's objective, without a clear connection between the two. While the introduction mentions vaccine hesitancy and acceptance, it does not explicitly identify a research gap that the study aims to address.

Suggestions

1. Streamline the introduction to focus on the study's objective and the specific context of university students in Enugu state, Nigeria.

2. Summarize or omit some statistics to maintain a clear and concise narrative.

3. Provide a clearer transition between the background information and the study's objective.

4. Identify a specific research gap that the study aims to address, and explain how the findings will contribute to the existing literature.

RESULT

The study achieved a good response rate, indicating a representative sample of the target population. The results provide a clear picture of the participants' demographic characteristics, such as age, gender, and admission route. The study assessed the participants' knowledge and perception of the COVID-19 vaccine, providing valuable insights into their understanding and attitudes. The study employed appropriate statistical tests (Chi-square) to determine associations between variables.

Although the response rate was high, the actual sample size (1,143) was lower than the estimated sample size (1,537), which might affect the precision of the results. The study only included students from three universities in Enugu State, Nigeria, which might not be representative of all university students in Nigeria. The study lacked a control group to compare the results with, which would have strengthened the findings.

I would recommend to Include additional variables, such as socioeconomic status, religious beliefs, and access to healthcare, to gain a more comprehensive understanding of the factors influencing vaccination decisions and to consider using more advanced statistical analysis techniques, such as logistic regression, to identify predictors of COVID-19 vaccine acceptance.

METHOD

The study is a cross-sectional survey with non-probability purposive and convenience sampling technique was used to obtain the data. Overall, the method section is technically sound, and the instruments and procedures are well-defined. However, some improvements could be made to increase the validity and generalizability of the study findings.

DISCUSSION

The discussion section appropriately interprets the findings, highlighting the level of knowledge, perception, and acceptance of the COVID-19 vaccine among undergraduate students and supports the findings with evidence from previous studies, demonstrating a good understanding of the topic.

The study acknowledges its limitations, such as the non-probability sampling technique and the limited generalizability of the findings and provides recommendations for future research, such as increasing the sample size and conducting pilot testing. However, the discussion section could have demonstrated more critical thinking, exploring potential biases and limitations in the study and include clearer implications for policy, practice, or future research. Also include the potential impact of the study's findings on public health initiatives and vaccination campaigns.

Reviewer #3: The topic is good and the work has been written well. Check Reference number 9. Analysis portion is good. Discussion has been written well. The findings and implications have been written well. The modal age category for this study was 18-24 years on what basis?

6. PLOS authors have the option to publish the peer review history of their article (what does this mean?). If published, this will include your full peer review and any attached files.

Reviewer #1: No

Reviewer #2: No

Reviewer #3: No

---

## [Author Response · Author response to Decision Letter 0]

31 Jul 2024

School of Medicine,

University of Nottingham,

Nottingham,

United Kingdom.

July 31, 2024.

The Editor-in-Chief,

PLOS ONE.

Dear Sir,

A point-by-point response to Reviewers’ comments

Please find below our response to the points raised by the reviewers.

Best wishes,

Blessing Onyinye Ukoha-Kalu; BPharm, MPharm, PharmD, PhD, FHEA

Title: “Factors associated with post-pandemic acceptance of COVID-19 vaccines among students in three Nigerian Universities”

RESPONSE TO REVIEWERS

REVIEWER 1

1. The authors need to review other local studies on this subject and situate them in the background to the study and discussion.

Thank you for your review and the suggestions of local studies to be included in the background and discussion. We have incorporated them on pages …

2. What constitutes good knowledge and perception needs to be outlined.

The respondents with scores equal or more than the median score have good knowledge while those having scores equal or above the median are categorized as having a positive perception. This statement has been inserted in the revised version of the manuscript.

3. The p-values in the tables should be placed in a separate column, not under the characteristics. Percentages of teh gender under knowledge added up to 1279%. It is not clear hwy this should be so. 

We have corrected the observations. Actually we made use of the percentages within each category of the variables to give a true picture of the proportions. 

4. Finally, the discussion needs depth. Results from this study and other referenced studies were merely quoted with little attempt to critically appraise the findings and situate them within the context of this study and referenced works. Feeble attempt was made to provide plausible explanation for most findings from this study.

The discussion has been rewritten considering your comments above. A critical appraisal and plausible explanation of most findings from the study were made within the context of the study and referenced works.

5. The authors have left out the main finding form this study, low COVID-19 uptake. Modifiable factors that affect this needs to be mentioned, as a way of recommendation. 

We have commented in the conclusion of the manuscript the need to modify the influencing factors that may increase the uptake of vaccine. 

6. Few edits and comments are made in the attached manuscript.

The edits were accepted while other minor corrections were effected in the revised version of the manuscript.

REVIEWER 2

Introduction

Suggestions

1. Streamline the introduction to focus on the study's objective and the specific context of university students in Enugu state, Nigeria.

Thank you for this suggestion. We have tried to focus more on the study’s objective.

2. Summarize or omit some statistics to maintain a clear and concise narrative.

We have expunged some statistics as well as made the narrative concise.

3. Provide a clearer transition between the background information and the study's objective.

We have provided a clearer transition between the background information and the study’s objective.

4. Identify a specific research gap that the study aims to address, and explain how the findings will contribute to the existing literature.

We have identified a specific research gap that the study aims to address and also explained how the findings will contribute to the existing literature.

Results

I would recommend to Include additional variables, such as socioeconomic status, religious beliefs, and access to healthcare, to gain a more comprehensive understanding of the factors influencing vaccination decisions and to consider using more advanced statistical analysis techniques, such as logistic regression, to identify predictors of COVID-19 vaccine acceptance.

Thank you very much for this suggestion. Unfortunately, we would not be able to incorporate them. However, we shall make reference to them under limitations of the study.

Method

some improvements could be made to increase the validity and generalizability of the study findings. 

Thank you for the suggestions. There was an error in the method section. We meant to say face validation and not content validation. This will be included in the limitations of the study. There was no content validation analysis. 

Discussion

However, the discussion section could have demonstrated more critical thinking, exploring potential biases and limitations in the study and include clearer implications for policy, practice, or future research. Also include the potential impact of the study's findings on public health initiatives and vaccination campaigns.

We have tried to write the discussion section demonstrating more critical thinking and exploring potential biases and limitations in the study as well as clearer implications for policy, practice or future research. 

REVIEWER 3

1. Check Reference number 9.

2. The modal age category for this study was 18-24 years on what basis?

1. Please, It is not clear what you wanted us to check about reference number 9.

2. The modal age category meant that majority of the respondents fell within this age range. However, since previous reviewers have queried the relevance of this to the objectives of the study, we have expunged it from the discussion section.

---

## [Editor Report · Decision Letter 1]

28 Aug 2024

PONE-D-24-14109R1Factors associated with post-pandemic acceptance of COVID-19 vaccines among students in three Nigerian UniversitiesPLOS ONE

Dear Dr. Ukoha-kalu,

Thank you for submitting the revised manuscript to PLOS ONE. We appreciate the efforts to address the previous concerns. However, after careful consideration, we feel that it has merit but does not fully meet PLOS ONE’s publication criteria as it currently stands. Therefore, we request further revisions to enhance the manuscript quality. 

Specifically, we ask you to:

1. Provide more justification for selecting three institutions from the possible 30 in Enugu State. Please clarify the sampling technique used and the selection criteria.

2. Revise the conclusion to focus on the most important results and their significance. Clearly state the study's contributions, implications, and provide a final thought or call to action (around 100-150 words).

3. Proofread the manuscript to correct spelling errors, ensure clarity and precision in your writing.

We look forward to receiving your revised manuscript.

Kind regards,

David Chibuike Ikwuka, Ph.D.

Academic Editor

PLOS ONE
---

## [Author Response · Author response to Decision Letter 1]

3 Sep 2024

The Editor-in-Chief,

PLOS ONE.

Dear Sir,

A point-by-point response to Reviewers’ comments

Please find below our response to the points raised by the reviewers.

Best wishes,

Blessing Onyinye Ukoha-Kalu; BPharm, MPharm, PharmD, PhD, FHEA

Title: “Factors associated with post-pandemic acceptance of COVID-19 vaccines among students in three Nigerian Universities”

RESPONSE TO REVIEWERS

REVIEWER 1

1. Provide more justification for selecting three institutions from the possible 30 in Enugu State. Please clarify the sampling technique used and the selection criteria.

Thank you for your comment. We have now provided more justification for selecting three institutions. Please see page 4:

“…These institutions were chosen based on their large student populations, diverse academic offerings, and significant geographical coverage within the state. We prioritized institutions with large and diverse student bodies to ensure that our sample would be representative of the broader student population in Enugu State. To capture a range of perspectives and experiences, we selected institutions that differ in their academic focus and type (a federal university, a state university, and a polytechnic). The selected institutions are strategically located in different parts of Enugu State, ensuring that the study captures data from students in various urban and rural settings…

… This study employed a non-probability purposive and convenience sampling technique to recruit the participants.”

2. Revise the conclusion to focus on the most important results and their significance. Clearly state the study's contributions, implications, and provide a final thought or call to action (around 100-150 words).

Thank you for your comment. We have now revised the conclusion to focus the most important results and their significance. Please see page 8:

“This study provides valuable insights into the post-pandemic acceptance of COVID-19 vaccines among university students in Enugu State, Nigeria. Despite a moderate level of knowledge and generally positive perceptions towards the vaccine, acceptance rates remained low, influenced significantly by the students' perceptions rather than their knowledge alone. The findings underscore the importance of targeted educational interventions that not only disseminate accurate information but also actively address misconceptions and enhance positive perceptions. This approach could be instrumental in improving vaccine uptake among this demographic. Our study highlights the critical role of perception in vaccine acceptance, suggesting that future public health campaigns should prioritize shifting perceptions through culturally sensitive, institution-specific strategies. These findings contribute to the ongoing dialogue on vaccine hesitancy and suggest actionable steps for increasing vaccine acceptance in similar contexts.”

3. Proofread the manuscript to correct spelling errors, ensure clarity and precision in your writing.

Thank you for your comment. We have now proofread the manuscript using an English editing service.

---

## [Editor Report · Decision Letter 2]

13 Sep 2024

Factors associated with post-pandemic acceptance of COVID-19 vaccines among students in three Nigerian Universities

PONE-D-24-14109R2

Dear Dr. Ukoha-kalu,

We’re pleased to inform you that your manuscript has been judged scientifically suitable for publication and will be formally accepted for publication once it meets all outstanding technical requirements.

Kind regards,

David Chibuike Ikwuka, Ph.D.

Academic Editor

PLOS ONE
---

## [Editor Report · Acceptance letter]

11 Oct 2024

PONE-D-24-14109R2 

PLOS ONE

Dear Dr. Ukoha-kalu, 

I'm pleased to inform you that your manuscript has been deemed suitable for publication in PLOS ONE. Congratulations! Your manuscript is now being handed over to our production team.

Kind regards, 

on behalf of

Dr David Chibuike Ikwuka 

Academic Editor

PLOS ONE